# 3D Cultures of Salivary Gland Cells in Native or Gelled Egg Yolk Plasma, Combined with Egg White and 3D-Printing of Gelled Egg Yolk Plasma

**DOI:** 10.3390/ma12213480

**Published:** 2019-10-24

**Authors:** André M. Charbonneau, Joseph M. Kinsella, Simon D. Tran

**Affiliations:** 1Faculty of Dentistry, Craniofacial Tissue Engineering and Stem Cells Laboratory, McGill University, Montréal, QC H3A 0C7, Canada; andre.charbonneau2@mail.mcgill.ca; 2Faculty of Engineering, Department of Bioengineering, McGill University, Montréal, QC H3A 0C7, Canada; joseph.kinsella@mcgill.ca

**Keywords:** 3D-Bioprinting, 3D-Printing, salivary glands, 3D-Cryo well insert, histology, egg yolk plasma, egg white, gel, Ki-67, rheology, cell culture, tissue engineering

## Abstract

For salivary gland (SG) tissue engineering, we cultured acinar NS-SV-AC cell line or primary SG fibroblasts for 14 days in avian egg yolk plasma (EYP). Media or egg white (EW) supplemented the cultures as they grew in 3D-Cryo histology well inserts. In the second half of this manuscript, we measured EYP’s freeze-thaw gelation and freeze-thaw induced gelled EYP (_G_EYP), and designed and tested further _G_EYP tissue engineering applications. With a 3D-Cryo well insert, we tested _G_EYP as a structural support for 3D cell culture or as a bio-ink for 3D-Bioprinting fluorescent cells. In non-printed EYP + EW or _G_EYP + EW cultures, sagittal sections of the cultures showed cells remaining above the well’s base. Ki-67 expression was lacking for fibroblasts, contrasting NS-SV-AC’s constant expression. Rheological viscoelastic measurements of _G_EYP at 37 °C on seven different freezing periods showed constant increase from 0 in mean storage and loss moduli, to 320 Pa and 120 Pa, respectively, after 30 days. We successfully 3D-printed _G_EYP with controlled geometries. We manually extruded _G_EYP bio-ink with fluorescence cells into a 3D-Cryo well insert and showed cell positioning. The 3D-Cryo well inserts reveal information on cells in EYP and we demonstrated _G_EYP cell culture and 3D-printing applications.

## 1. Introduction

Salivary hypofunction can be induced by side-effects of medicines, Sjogren’s autoimmune disorder, or head and neck radiation therapy. This condition is estimated to affect 20–30% of the adult population [1]. Since no permanent salivary restoration treatment exists, the engineering of miniature salivary secretory units could improve a patient’s quality of life. For tissue engineering, scientists rely on synthetic and natural biomaterials [2] (such as protein and carbohydrates) and a decellularized extracellular matrix, which enhance the growth environment. Scientists have used singular- [3,4,5,6,7] and multi-compositional [8,9,10,11] biomaterials in an attempt to engineer human salivary glands (SGs). In the tissue engineering of soft tissues, gel-like biomaterials are known to improve cell distribution. 3D distribution in biomaterials offer better recapitulation of the native salivary cell tissue mechanical supporting environment and promote tissue development [12,13]. Furthermore, 3D-Bioprinting gels can improve 3D cultures, since they permit researchers to precisely place clusters of cells in specific locations [14]—the location of cells impacts morphogenesis and patterning [15]. For gel-like biomaterials, extrusion-based 3D-printing is most suitable [16]. In our search for a bio-inspired, cost-effective, multi-compositional soft biomaterial, we recently discovered that a translucent fraction from the egg yolk (EY)—the egg yolk plasma (EYP)—can under certain circumstances permit media-free human cell survival. This current study builds on our previous data as we continue developing and understanding the EYP biomaterial for soft tissue engineering.

The egg continuously produces entire living organisms. Natural selection optimizes the avian egg to produce tissues ex vivo. Other scientists have also caught on to this concept and used egg biomaterials to host human cells. Human cells have been added to developing fertilized chick embryo [17,18], egg white (EW) [19,20], and EY [21,22,23]. Additionally, pharmacological studies use the egg environment for drug screening [24]. In another of our reports on egg biomaterials [Submitted], we also exposed two salivary cell types to EYP + Media or EYP + EW. In that report, only live/dead stains examined the cells’ survival conditions over 14 days. Under certain conditions, we discovered how human cells can survive and/or expand without cell culture media. Additional data from other angles of view and techniques could provide us with more information on human cell behaviour in egg-derived biomaterials. As for EYP’s gelation for 3D cultures or 3D-Bioprinting applications, several food science reports have investigated EY and EYP’s gelation [25,26,27,28,29,30,31] and proposed theories on the mechanism of freeze-thaw gelation [25,27,28,29,30,31,32]. To create gelled EYP (_G_EYP), the literature has reported optimal freeze-thaw gelation of EYP between −12 [28] and −21 °C [32]. Furthermore, the longer EYP freezing time, the stiffer the gel becomes once thawed to 25 °C. No study has specifically investigated _G_EYP’s viscoelastic properties at 37 °C, the physiological temperature. Furthermore, 3D-Printing or 3D-Bioprinting of egg-derived materials for tissue engineering purposes has never been attempted. Slightly different from standard 3D-Printing, bioprinting involves cell-laden biomaterials (bio-inks). Scientists can use imaging technologies to scan and reproduce tissues [33]. To claim basic 3D-Printing, it requires demonstrating biomaterial extrusion, structural maintenance and statements on appropriate printing pressure, layer separation, layer height, and printing nuzzle speed [34]. In addition, it is important to investigate the printed structure’s swelling, porosity, and degradation [35].

Here, we used several experiments to examine human cells’ behaviour in egg biomaterial combinations and test other egg biomaterial applications for tissue engineering. More specifically, we first hypothesized that our well insert technology [36] would provide more information on cell distribution and proliferative state. Both human SG cell types (acinar [37,38] and stromal) were cultured in similar EYP + Media and EYP + EW in the 3D-Cryo well insert and stained with Sirius Red [39] and Ki-67 [40] at 0 and 14 days. In addition to the EYP + EW gel-like biomaterial, we hypothesized that a freeze-thawed EYP (now _G_EYP) could still possess gel-like properties, if brought to 37 °C (relevant for cell culture applications). To test this hypothesis, we used a rheometer to examine _G_EYP’s viscoelastic properties at 37 °C after freeze thaw treatment. Rheometers can test basic viscoelastic material characteristics. These mainly include the stress stored in the material (storage or elastic modulus G’ in Pa) and the stress energy lost from dissipation (loss or viscous modulus G” in Pa) [41]. We used the formed _G_EYP in two subsequent experiments to test its 3D culture and 3D-Bioprinting potential. As a 3D culture model, we used _G_EYP in combination with EW to reproduce an interface similar to EY + EW in eggs [42]. We called this the “Interface model”. Cultured in the 3D-Cryo well insert, we hypothesized that the _G_EYP + EW interface system could be a stand-alone tissue engineering incubator requiring no media. We used histology to evaluate over 14 days how _G_EYP stiffness maintained NS-SV-AC cells at the interface and their Ki-67 expression. For 3D-Bioprinting, we hypothesized that we could 3D-Bioprint two inks into a 3D-Cryo well insert and visualize the two cell types interface on a sagittal section. We called this 3D-printed model “the ball and socket”. The two different bio-inks simulate the epithelial parenchymal to mesenchymal stromal interface, both initiators of tubules and acini during development [15]. 

## 2. Materials and Methods

### 2.1. Egg Yolk Plasma (EYP) or Egg White (EW) Isolation and Heat Treatment

We purchased fresh eggs (Large White Eggs Omega-3, President’s Choice) from a local retail store (Montreal, QC, Canada). With the apex of the shells cracked, the EW was poured into a sterilized 250 mL beaker while the yolk remained in the shell. Thereafter, the EY was gently poured onto an absorbent paper. On the paper, the EY was rolled and transferred to another sterile 250 mL beaker. Similarly, 4 yolks were processed and pooled in the beaker, then gently mixed with a spatula; the contents of this beaker were split into numerous 2 mL tubes. We centrifuged the 2 mL tubes samples at 15,400 RCF, 25 °C for 6 h. After centrifugation, the translucent supernatants were pooled into a 50 mL tube and stored at 4 °C until use. For the EW, they were pooled and then minced by filtration. The mincing involved vacuum suctioning the EW through a 1–2 mm sized porous filter (60240, Coorstek, Golden, CO, USA) three times. EW was also stored at 4 °C in a 50 mL tube. Prior to use with cells, both 50 mL tubes containing either EYP or EW were placed in a 60 °C water bath (“pasteurization”). When the internal egg biomaterial’s temperature attained 55–60°C for at least 5 min, the samples were rapidly cooled in a −80 °C freezer for 20 min. After freezing, they were incubated at 37 °C in a water bath or stored at 4 °C for long-term storage.

### 2.2. Well Insert Technology Design and Fabrication

Previously, we described in detail the design and fabrication of the 3D-Cryo well insert [36]. Briefly, gelatine pills (size 000, Capsuline ®, Pompano Beach, FL, USA) were coated twice with paraffin (or Histowax) (00403, HistoLab, Espoo, Finland) heated to liquid, cooled, and exposed 20 min to UV lights. The capsules’ quality control involved incubation in standard cell culture conditions overnight with media. 

### 2.3. HuSG-Fibro Isolation and Standard Culture of Both HuSG-Fibro and NS-SV-AC

The isolation followed the alternative protocol in Current Protocols in Cell Biology named, “Establishment of fibroblast cultures” [43]. Briefly, we minced tissues with scalpels and incubated the tissues with 1000 U/mL of collagenase Type 1 (LS004196, Worthington Biochemical Corp., Lakewood, CA, USA) at 37 °C for 2 hours. We grew cells in complete growth media composed of RPMI 1640 (21870-076, Thermo-Fischer Scientific, Grand Island, NE, USA), 10% FBS (12483020, Thermo-Fischer Scientific, Burlington, ON, Canada), 1% 1 M HEPES (H3375, Sigma-Aldrich, St-Louis, MO, USA), 1% non-essential amino acids (11140050, Thermo-Fischer Scientific, Burlington, ON, Canada), 1% L-glutamine (25030081, Thermo-Fischer Scientific, ON, Canada), 1% penicillin/streptomycin (10,000 U/mL) (15140122, Thermo-Fischer Scientific, Burlington, ON, Canada), 1% sodium pyruvate (11360070, Thermo-Fischer Scientific, Burlington, ON, Canada). The cells attached overnight and over the next 2–3 days a media replacement washed the culture and removed non-adherent cells and erythrocytes. We expanded NS-SV-AC in EpiMax Media (002010024 CL, Wisent Bioproducts, St-Bruno, QC, Canada). The EpiMax contained supplements: hormones and bovine pituitary extract (002013024, Wisent), anti-fungal supplement amphotericine B (002014016 IL, Wisent Bioproducts, St-Bruno, QC, Canada), and antibiotic supplement gentamicin (002015017 TL, Wisent Bioproducts, St-Bruno, QC, Canada). Expansion occurred in either plastic 6 well plates or 100 mm dishes. We passaged both cell types with 0.05% Trypsin/EDTA (25200072, Thermo-Fischer Scientific, Grand Island, NE, USA) and re-suspended in smallest possible volume prior to adding to biomaterial <500 μL. We briefly characterized both cell types grown on glass with Ki-67 (Appendix A). 

### 2.4. Cell Culture in 3D-Cryo Well Insert Technology

Briefly, cells were passaged, counted and directly mixed at a concentration of 225 × 10^3^ cell/100 μL in each biomaterial. To the initial 100 μL/well and at every 4th day, 50 μL of biomaterial mixtures without cells were added to provide nutrition and hydration to the cells. For each biomaterial, 3 to 8 3D-Cryo well inserts were frozen and sectioned for each time point (0 and 14 days). Two cell-less controls were frozen and cut. See our previous publication for more details on this technique [36].

### 2.5. Cryosectioning, Chemical Staining, and Immunohistochemistry Protocols

The 3D-Cryo well inserts were removed with tweezers from 48 well culture plates and frozen in liquid nitrogen (N_2(Liq)_). After this first freezing, a secondary freezing step occurred to embed the well insert in OCT 37% (diluted in water) (1708801018, Algol Diagnostics Oy, Espoo, Finland). As a result of this secondary freezing, the 3D-Cryo well insert was preserved in an ice cylinder. To perform this secondary embedding, the following steps were performed: a cylindrical mould was created from the lid and 3 cm long section of a 15 mL tube (the mark below the 12 mL was cut off). In the plastic cylindrical mould, OCT 37% was poured. The OCT containing plastic mould held by tweezers was incubated into N_2(Liq)_. It was maintained in the N_2(Liq)_ only until the OCT on the inner edges of the plastic mould froze (~1 min). At this point, the plastic mould removed from the N_2(Liq)_ left just enough place for the insertion of the 3D-Cryo well insert. Placing everything back in the N_2(Liq)_ completed the second freezing (embedding) step. After freezing, the OCT cylinder—containing the 3D-Cryo well insert—was left to thaw 5 min. After the thaw, when frost accumulated on the outer edges of the plastic tube, the OCT cylinder containing the 3D-Cryo well insert was ejected. This cylinder containing the sample was then cut on a cryotome at 14–20 μm thicknesses and sections mounted on glass slides. After air drying, the samples were stored at −20 °C or processed for 10 min in homemade 10% Neural Buffered Formalin (F1635, Millipore Sigma, St-Louis, MO, USA), then 10 min in 1% Triton X-100 (T8787-50, Millipore Sigma, St-Louis, MO, USA) for cell permeation. The fixed and permeated tissues were then stained. For Sirius Red, the slides were first counter stained in Weighert’s Hematoxylin composed of hematoxylin (4305, Merck, Darmstadt, Germany) and Iron (III)-chloride hexahydrate (Usually anhydrous; therefore, in our case—without anhydrous, molar equivalents were calculated.) (3943, Merck, Darmstadt, Germany) [44] for 10 min. From the hematoxylin, the slides were individually gently stirred and stored in water. Then, the slides were exposed for maximum of 3 min to homemade Sirius Red (0.1 g of Direct Red 80 (365548-5G, Millipore Sigma, St-Louis, MO, USA) in 100 mL of saturated picric acid solution (P6744, Millipore Sigma, St-Louis, MO, USA). The slides were then exposed for 5 min in 5% acetic acid in water, transferred to 100% ethanol, completely dehydrated with xylene, and mounted with organic media (03989, Fluka Analytical, Seelze, Germany). For the immunohistochemistry, controls included human brain, myoma tissue, and mouse tongue. (Appendix A). First Ki-67 (Clone MIB-1, 80 mg/mL, Dako, Glostrup, Denmark or D3B5, Cell Signalling Technology Inc., Danvers, MA, USA) at 1:125 in PBS was used with an automatic slide stainer (BOND-MAX, Leica Biosystems, Nussloch, Germany). We treated the slides with bond polymer refined detection kit (DS9800, Leica Biosystems, Nussloch, Germany). This involved incubation with the primary antibody for 60 min, 30 min in the post primary, 30 min in the polymer, 10 min in the 3,3’-Diaminobenzidine (DAB), and 5 min in the hematoxylin. The slides were washed with the washing buffer provided by the company. 

### 2.6. Gelation of Egg Yolk Plasma (EYP) and Biomaterial Testing

After heating and freezing treatment (“pasteurization”) or storage at 4 °C, the EYP samples were stabilized to room temperature and directly frozen in a −20 °C freezer. Glass 10c.c (512142, BD Bioscience, Franklin Lakes, NJ, USA) or plastic (7012114, Nordson Corporation, Westlake, OH, USA) syringes sealed with its piston and paraffin paper were used during freezing containing 3–7 mL of EYP. After freezing, the samples were heated to 37 °C or placed in 4 °C for long-term storage until rheology testing. Prior to testing, the _G_EYP samples were preheated for 2 hours at 37 °C in an air incubator. On the rheometer (MCR 302, Anton Paar, Montreal, QC, Canada), the time test parameters were: frequency 0.1 Hz, shear strain 0.1%, temperature 37 °C, with a 1 mm gap, using 25 mm parallel plate geometry (PP25, Anton Paar, Montreal, QC, Canada). A mineral oil was used to seal the gap between geometry and Peltier element (base plate). Each sample was tested every minute for 120 min. For rheology, we implemented a three-month cutoff from gelation to test date. Accordingly, n values accounted for 12–36 samples per time point. A yield stress experiment was conducted with similar parameters. The shear stress increased by 2 Pa/s from 1–500 Pa and n-values were T = 0, n = 6 and T = 30 d, n = 4.

### 2.7. NS-SV-AC in Gelled EYP (_G_EYP) + Egg White (EW) Cell Interface Model 

In a 3D-Cryo well insert, we deposited three layers First 200 μL of 30 day _G_EYP, then 2.50 × 10^5^ NS-SV-AC cells in ~70 μL of media, then 200 μL of minced EW. These were incubated in at 48 well plate without additional media or biomaterial for 14 days. Three 3D-Cryo well insert samples were flash frozen at both 0 and 14 days. One cell-less control was frozen and cut.

### 2.8. 3D-Printing the ball and socket, Speeds and Nuzzles

_G_EYP was 3D-printed with GeSiM bio-printer (Bioscafolder 3.1, GeSiM, Radeberg, Germany) at 25 °C. Viscoelastic properties tested at 37 °C before printing were _G_EYP: G’ = 693 G” = 206. With 10 cc cartridge (7012114, Nordson Corporation, Westlake, OH, USA), printing pressures were of 200 kPa, needle size 23g—stainless steel precision tip (7018305, Nordson Corporation, Westlake, OH, USA), layer separation equalled 400 μm/layer and 2 mm/s for printing speed. For colouring _G_EYP, one drop of food colouring was added to _G_EYP and mixed by manual piston activation (Appendix A).

### 2.9. Addition of Cells to Gelled EYP (_G_EYP) and Manual Extrusion

NS-SV-AC cells were split into two 100 mm dishes each with 5 mL of PBS and fluorescent dyes. For green fluorophore, cells were labelled with CFSE (Ab113853, Abcam, Cambridge, UK) at 0.01 mM final concentration. For blue fluorophore, cells were labelled with Hoeschst (33342, Thermo Fisher Scientific, Carlsbad, CA, USA) at final concentration of 0.022 μg/mL. In 3 mL of _G_EYP from the 3D-printing experiment, 2.50 × 10^5^ labelled green or blue cells were added with 500 μL of green or red food coloured PBS. A metal stopcock link connected the first syringe containing either coloured cells and gels to the base of a second empty syringe (Appendix A). Through 60 slow piston actions, the cells and the dye distributed themselves into each respective gel. The complete distribution of the cells was estimated based on the distribution of the dye. Manual extrusion into the 3D-Cryo Well Insert was performed with 20 g precision stainless steel tip (7018166, Nordson Corporation, Westlake, OH, USA).

### 2.10. Imaging

Images were captured with ×10 or ×20 objectives using either DM4000 microscope (Leica Biosystems, Nussloch, Germany) or Axio Imager (Carl Zeiss MicroImaging GmbH, Gottingen, Germany) equipped with or without fluorescent filters.

### 2.11. Statistical Analysis

GraphPad Prism version 7 (GraphPad Software, San Diego, CA, USA) was used to perform the statistical analysis. Data were presented with mean ± S.E. and when appropriate were analyzed by one-way ANOVA followed by Tukey’s Post-Hoc test. The statistical significance was defined as p-value < 0.05.

## 3. Results

We provide a summary of the figures in Table 1.

### 3.1. NS-SV-AC Cells’ Distribution in Egg Yolk Plasma (EYP) Mixtures

At both 0 and 14 days, Sirius Red stained biomaterial mixtures were intensely red and nuclei black (Figure 1). In the 3D-Cryo well inserts, the cells in EYP + Media biomaterial—without structural support, rapidly sank (Figure 1A). In contrast, EYP + EW improved cell distribution. In EYP + EW, we noted vertical distribution of the cells at T = 0. Later, at T = 14d, the EW network sagged but maintained better cells’ vertical distribution than EYP + Media. Near the base of the well, the EW fibres often accumulated in a distinguishable band of fibres (Figure 1B). In EYP + EW, individually distributed cells at T = 0 enabled us to observe the formation of small cell clusters at T = 14 days. Surprisingly after 14 days in EYP + EW, the cells’ size appeared smaller than in EYP + Media.

### 3.2. NS-SV-AC’s and HuSG-Fibro’s Ki-67 Expression in Egg Yolk Plasma (EYP) + Media or EYP + Egg White (EW)

We used IHC to detect the cells’ Ki-67 expression (a marker of proliferating cells) when grown in biomaterials (Figure 2) and in controls (Appendix A). For cancer tissue—human leiomyoma seeded with cancer cells—we observed a few positive dark brown nuclei (Appendix A); mouse tongue epithelium and tumour also had positive nuclei (Appendix A). When we grew cells on glass, the NS-SV-AC’s nuclei expressed Ki-67, while a minority of HuSG-Fibro’s nuclei marked positive (Appendix A). With biomaterials EYP + Media, EYP + EW or media (control) in the well inserts, NS-SV-AC stained positively to Ki-67 at all times and conditions (Figure 2A). NS-SV-AC grown in egg-derived biomaterials expressed Ki-67 similar to growth in media (serum free EpiMax). For the HuSG-Fibro at T = 0, Ki-67 expression marked very few cells. After 14 days, positive Ki-67 signal limited itself to cells grown in media (Figure 2B). HuSG-Fibro grown in egg-derived biomaterials expressed less Ki-67, as compared to when grown in cell culture media (RPMI 1640 + 10% FBS). In all biomaterial combinations and media, NS-SV-AC expressed more Ki-67 than HuSG-Fibro.

### 3.3. Freeze and 37 °C Thaw Gelation of Egg Yolk Plasma (EYP) 

A freeze of −20 °C and thaw of 37 °C formed on an EYP gel (Figure 3). Over the 30 days tested, approximately each day of freezing increased elastic modulus (G’) by 10 Pa (Figure 3A). We observed the first significant difference after seven days of freezing. Although a noticeable clear trend is seen, variability existed; in Figure 3, we show standard error. Standard deviation of storage modulus (G’) on days 0, 7, 15, and 30 were 0.82, 76.1, 81.8, and 115.3 Pa, respectively. Importantly, freeze and thaw gelation can form a gel even at 37 °C (Figure 3B). 

### 3.4. Gelled EYP (_G_EYP) + Egg White (EW) Cell Interface Culture

With the ability to produce a _G_EYP, we considered _G_EYP’s ability to support cells at an interface with EW. This would somewhat reproduce EY and EW’s interface seen in the chicken’s egg (Figure 4A,B). We produced this interface in a 3D-Cryo well insert and seeded NS-SV-AC cells. After the addition of the final EW layer, macroscopically, the EW did not appear to mix with the _G_EYP; an interface was maintained (Figure 4B). Resulting cryosections and IHC staining of this interface culture showed how the interface could be maintained for at least 14 days (experimental endpoint) (Figure 4D). Over 14 days, the _G_EYP did not appear to lose its viscoelastic properties. In contrast, histology demonstrated EW’s compaction losing its diffuse T = 0 appearance. By IHC using anti-Ki-67 antibody, the NS-SV-AC’s had an omnipresent nuclear expression at T = 0 (Figure 4C). Later at 14 days, the expression visually appeared to decrease by at least 50% (Figure 4D). 

### 3.5. Gelled EYP (_G_EYP)’s Potential for 3D-printing Tissues

3D printers can extrude biomaterials with and without cells. Here, we first tested _G_EYP’s printing potential without cells. Pneumatic pressures—200–300 kPa—extruded _G_EYP that had viscoelastic values G’ = 693 Pa and G” = 206 Pa. The design cylinder had a layer spacing of 400 um/layer, extruded well at speeds of 2 mm/s, and was tested to 5 mm in height (Figure 5A). After printing, exposure to room temperature air gradually shrunk the cylinders (Appendix A); importantly, sealing the printed structured mitigated the shrinking (Appendix A). When exposed to media, the media degraded the culture after the second rinse on day 3 (Appendix A). With only the addition of media—and not removal, gels maintained cells in 3D distribution for at least 17 days (Appendix A). The 3D printer also printed two different _G_EYP inks on the same structure (Figure 5B). For this double cartridge structure “the ball and socket”, we used similar printing parameters. Next, we added cells to _G_EYP. When we added cells to the biomaterials by manual mixing (Appendix A), the viscoelastic properties visually decreased and required printing pressure dropped to 100 kPa (data not shown). To observe the cells’ positioning within the ball and socket, we manually extruded two cell-laden biomaterials into the 3D-Cryo well insert. Microscopic analyses of cryo-cut sagittal sections showed that both fluorescent cell types (green and blue) maintained their extruded green and red bio-ink interface and proximity (Figure 5C). 

## 4. Discussion

In this study, we completed additional experiments to further explore EYP as a biomaterial for 3D Cultures and 3D-Printing experiments. As a tool for analysis, we used our recently developed 3D-Cryo well insert to examine by histology the behaviour of cells in various EYP mixtures. Importantly by histology, we showed a never seen before perspective of cells in EYP, EYP + EW, _G_EYP + EW and in manually extruded _G_EYP.

In histology, scientists use Sirius Red to locate collagens in a tissue [39]. Although scientists use it for this application, other reports mention that it stains non-collagenous structures [39,45,46,47,48]. To our knowledge, our observations are the first to report Sirius Red’s bright red staining of egg biomaterials. The Sirius red’s rapid vivid stain did not provide information on cells’ collagen synthesis as we expected. Rather, because of the good contrasts it provided between cells and biomaterials, we now used this chemical stain to identify the cells location in the biomaterial. EYP + Media could not retain the cells from sinking. In contrast, the EYP + EW’s mixture maintained the cells above the well’s base. The vacuum funnel’s mincing—during the EW’s preparation—broke down the EW’s larger gelatinous structure into small gel particles. The small EW particles, only existing in the EYP + EW mixture, maintained the cells suspended in 3D. Here, as we discuss our attempts to stain cell’s collagen production, we feel it is important to discuss a supplementary IHC experiment (Appendix A) we conducted on cells’ collagen production. Over 14 days, we did not observe more collagen expression with both cell types in EYP + Media or EYP + EW. Although the vendor advertised the antibody as a collagen panel antibody (I-V), our controls rather stained in majority a collagen from smooth muscle. This is supported by IHC positivity in high smooth muscle tissue controls human leiyomayoma (Appendix A) and longitudinal blood vessels in mouse tongue (Appendix A). Against expectations, no collagen was seen in the mouse’s tongue basement membrane (Appendix A), and human SGs (Appendix A). In future studies, we recommend other collagen antibodies or detection of other extracellular matrix proteins (laminin [49]).

IHC also evaluated cells’ Ki-67 expression. In human SG tissues, Ki-67’s values range from medium to none [50]; therefore, scientists do not have a confident in-vivo image of Ki-67’s SG expression. NS-SV-AC’s generalized Ki-67 expression follows the observations from our previous 3D-Cryo well insert study [36]. In that study, we attribute the omnipresent Ki-67’s expression to the SV-40 virus used for the cell line’s immortalization [51]. Although we noted a continuous Ki-67 expression in EYP + Media and EYP + EW (Figure 2A), they contrasted with results obtained from _G_EYP + EW interface experiment (Figure 4C,D). Ki-67 expression in EYP + Media and EYP + EW strongly suggests normal active NS-SV-AC cell proliferation. HuSG-Fibro’s near absent Ki-67 expression in both EYP + Media and EYP + EW models suggests our egg mixtures do not support rapid fibroblast proliferation. This slow proliferation result resembles another live/dead stain experiments in another manuscript on this cell types in these biomaterials showing survival but near senescence. In the _G_EYP + EW interface experiment, NS-SV-AC cells lost about 50% of their Ki-67 expression at 14 d suggesting suboptimal performance. Water evaporation over time from the EW compartments most likely led to the EW’s condensed appearance and decrease of Ki-67 expression at day 14. Seeing as stable humidity levels contribute to successful embryonic development [52,53] water addition or maintenance of the culture in a humid environment could maintain the system’s hydration and Ki-67 expression. Still, it remains uncertain if these non-ki-67 expressing cells at the interface were senescent—like in the HuSG-Fibro—or dead.

For EYP’s gelation, we selected freeze-thaw gelation over other methods [21,54,55] because of its ease; most labs possess −20 °C freezers. In line with our systems physiological needs, the rheometer conducted the viscoelastic tests at 37 °C, contrasting other food science studies [25,26,27,28,29,30,31]. Alternative to shear, compression tests have often measured soft tissue stiffness values and identify a tissue’s tensile storage modulus E’ (in Pa). In our shear rheology tests, an equation can approximately convert shear storage modulus G’ (in Pa) to E’. With the appropriate Poisson’s Coefficient (ν) and assuming the materials are isotropic and homogenous, the equation E = 2(1 + ν)G can approximately interchange the values; therefore, G’ is 2–3 times smaller. From some studies on glandular soft tissues, we note the following E’: healthy human breast 2800 (±1400) Pa [56], developmentally permissive salivary gland tissue 4000 Pa [12], mouse salivary glands in development (130 Pa), or adult mouse SG (2000Pa) [13]. Converting these values to G’ equal, respectively, 933 (±467) Pa, 1333 Pa, 43 Pa or 666 Pa. In our results, after 30 days freeze thaw treatment, our mean G’ was ~350 Pa (Figure 3). Importantly, EYP’s mean storage modulus lies within the same order of magnitude as glandular soft tissues. Absence of either cells or ECM components could explain the slight difference between the material’s storage modulus values. Over the freezing times, we measured a clear increase in the cell-less _G_EYP viscoelastic properties. Unfortunately, there existed considerable sample variability. Numerous studies [26,57] speak about various factors affecting the gelation of the EYP. We attempted to standardize many variables such as: 1) EYP homogenization time, 2) EYP pasteurization time and temperature, 3) EYP pre-freeze temperature, 4) EYP thaw temperature, 5) EYP freezing compartment material and size; and 6) sample preheating time. We also encourage others to control these variables. In our case, most often, the first sample of each syringe yielded the highest viscoelastic values, suggesting non-uniform gelation. Future studies should seek to standardize gelation for better freeze-thaw EYP gel’s reproducibility. 

In our experiments, we considered two _G_EYP applications; either as a support for growing cells at the interface between EYP and EW or, as a 3D-Bioprint able bio-ink. First, with overlay technique, the _G_EYP’s stiffness enabled maintenance of the cell at the _G_EYP + EW interface for 14 days (Figure 4). Three points can highlight this interface experiment’s significance. First, it demonstrates the _G_EYP’s ability to maintain cells at an interface between the EY and EW similar to avian embryonic development. Secondly, the absence of cell culture media did not entirely deplete the cell’s Ki-67 expression. Lastly, the 3D-Cryo well insert effectively demonstrates the internal appearance of the cells’ position in relation to the biomaterial; a perspective unachievable without the well insert technology. We next tested _G_EYP’s second hypothesized application, bioprinting. Without cells, the biomaterial extruded smoothly, held its own weight, and produced relatively complex 3D-printed structures (Figure 5A,B). When we incubated the cell-less printed structures in incubating conditions, the cylinders could still be distinguished for at least three days in media or PBS (Appendix A). After 10 days though, the printed _G_EYP structure could not be distinguished. With freeze-thaw _G_EYP, media changes are not compatible for long-term incubations. Interestingly, there may not be media requirement for _G_EYP cell culture. Our other submitted study shows that cells can survive in EYP without cell culture media. Essentially, media-free survival simply requires NaOH to modify EYP’s pH to 7.4 before gelation. EYP’s gelation with a modified pH has previously been discussed for food science applications [32]. To test this hypothesis, we showed freeze and thaw technique’s ability to still create a _G_EYP with a NaOH pH modified EYP at 37 °C (Appendix A). With the pH modified gel, cell embedding is the next challenge. In this study, we explore two methods to embed cells and in this discussion, we hypothesize a third. First we tried the double syringe mixing process (Appendix A), where it created bubbles causing an apparent decrease in the gel-like properties of the _G_EYP. Secondly, to decrease bubble formation, a metal spatula effectively mixed a small concentrated drop of cells in _G_EYP in a 30 mm dish (Appendix A). Here, the third untested hypothesis relies on EYP’s ability to act as cell cryoprotectants [58]. We hypothesize that freeze thawing a cell laden liquid EYP pH modified with NaOH to pH 7.4 could produce a _G_EYP bio-ink. Accordingly, freezing would induce the gelation of the cell-laden liquid EYP (pH 7.4) and when thawed, the _G_EYP pH 7.4 with cells could be bio-printed from the same syringe used for gelation.

## 5. Conclusions

The data supports the Cryo well insert as a tool to characterize cells in dense multi-compositional biomaterials (EYP + Media, EYP + EW and _G_EYP + EW) over time. Furthermore, Ki-67 stain confirmed NS-SV-AC cell’s proliferative potential and HuSG-Fibro’s relatively senescent state. In the second half of the manuscript, we fine-tuned EYP’s mechanical properties through freeze-thaw gelation and tested _G_EYP’s cell culture and 3D-printing potential. Our results showed that _G_EYP gelled for 30 days, and served as a sufficient mechanical support to maintain cells above the well’s base for 14 days. Interestingly in this media-free cell culture, the _G_EYP + EW interface allowed at least 50% of NS-SV-AC cells to remain proliferative. In the last experiments, we could not synthesize a bioprintable _G_EYP. In fact, _G_EYP was printable but not bio-printing (i.e. with cells already embedded in the gel), and would still require additional optimization steps. Unfortunately, as we added cells, bubbles formed and the mechanical properties appeared to change. Although we were not entirely successful with bioprinting cell-laden egg biomaterials, we remain optimistic that this combination holds important potential for human tissue engineering. The development of the egg components as a cost-effective alternative biomaterial makes a crucial contribution for scientists seeking a developmentally encouraging, multi-compositional, biomaterial for soft tissue engineering.

## Figures and Tables

**Figure 1 materials-12-03480-f001:**
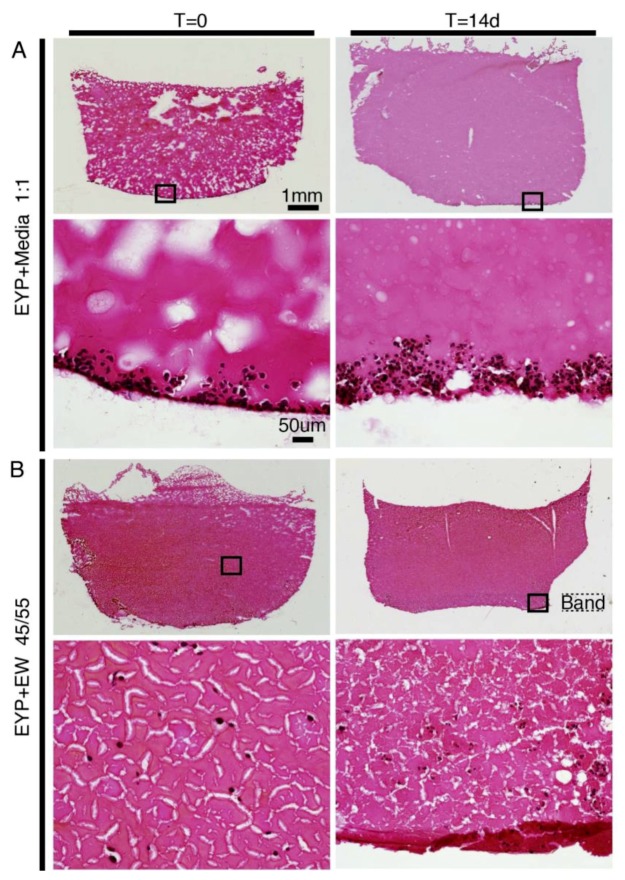
NS-SV-AC cells in EYP Biomaterial combinations analyzed at 0 and 14 days. In panels of biomaterial combinations, either EYP + Media or EYP + EW, images taken by microscopy show 25× (upper panel) and 200× (lower panel) magnification of the sagittal sections grown in 3D-Cryo well inset. Sirius Red Chemical Stain was used to stain sagittal sections. Biomaterials were in red/pink, while cells were in black. (**A**) Cells in EYP + Media 1:1 mixture; (**B**) Cells in EYP + EW 45:55 mixture. The word “band” highlights the approximate thickness of the sagged protein network in the EW at T = 14 d.

**Figure 2 materials-12-03480-f002:**
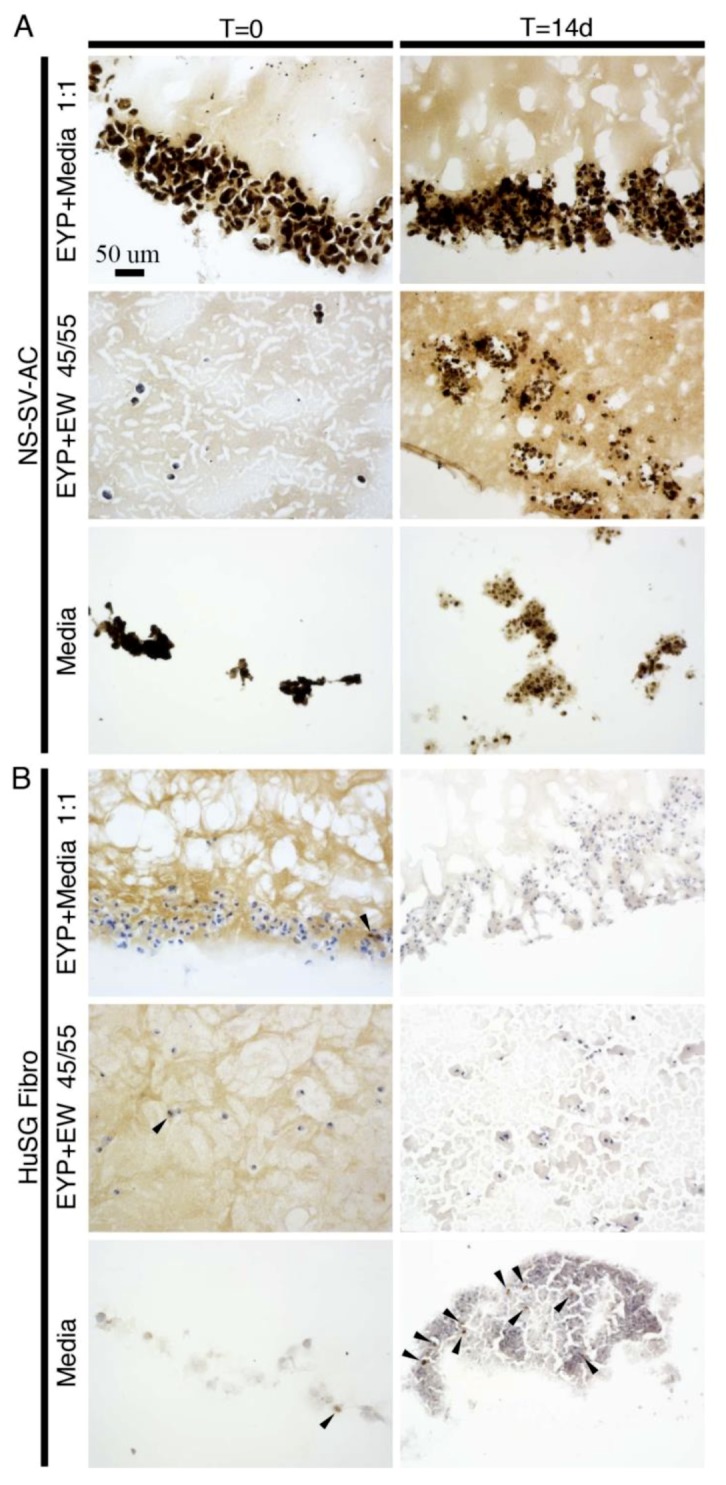
NS-SV-AC and HuSG Fibro cells in EYP biomaterial combinations and media analyzed at times 0 and 14 days with Ki-67 antibody using immunohistochemistry. Tissue sections were obtained from sectioning 3D-Cryo well inserts. Images were captured by microscopy with 20× objective. Biomaterials commonly appeared light brown, while the nuclei were blue. Presence of Ki-67 is shown in dark brown. With NS-SV-AC, the dark brown signal appeared to overlap blue nuclei colour. (**A**) NS-SV-AC cells in the biomaterials and media; (**B**) HuSG Fibro cells in the biomaterials and media. NS-SV-AC’s T = 0 Ki-67 in media image was reproduced with Wiley’s Biotechnology Journal’s permission from our own article. The article from Charbonneau et al. was titled “3D Culture Histology Cryosectioned Well Insert Technology Preserves the Structural Relationship between Cells and Biomaterials for Time-Lapse Analysis of 3D cultures” © 2019 WILEY-VCH Verlag GmbH & Co. KGaA, Weinheim.

**Figure 3 materials-12-03480-f003:**
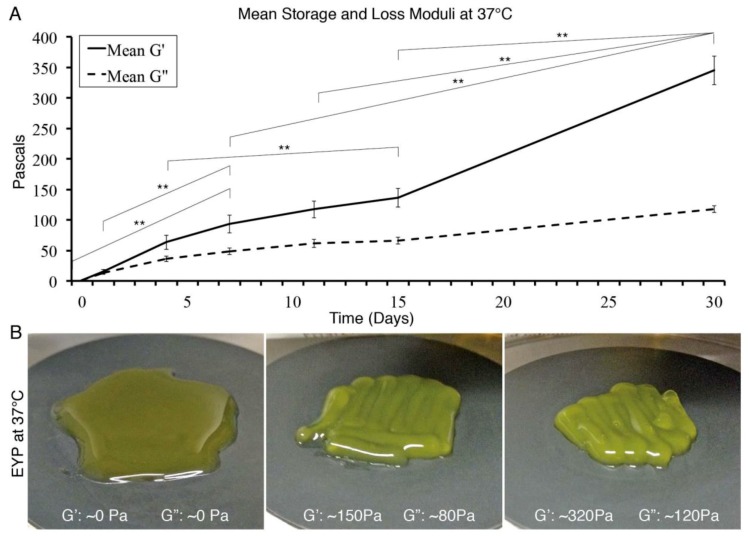
Viscoelastic properties of freeze thawed gelled EYP measured at 37 °C, 0.1Hz, and 0.1%. Error bars show the standard error and statistical significance p-value < 0.01 only shown for G’. Statistical significance was calculated using one-way ANOVA, followed by Tukey’s Post-Hoc test. (**A**) _G_EYP’s viscoelastic properties after freeze thaw treatment; (**B**) Images of EYP at 37 °C without freeze thaw (left), freeze thaw treatments for 15 days (centre) and 30 days (right).

**Figure 4 materials-12-03480-f004:**
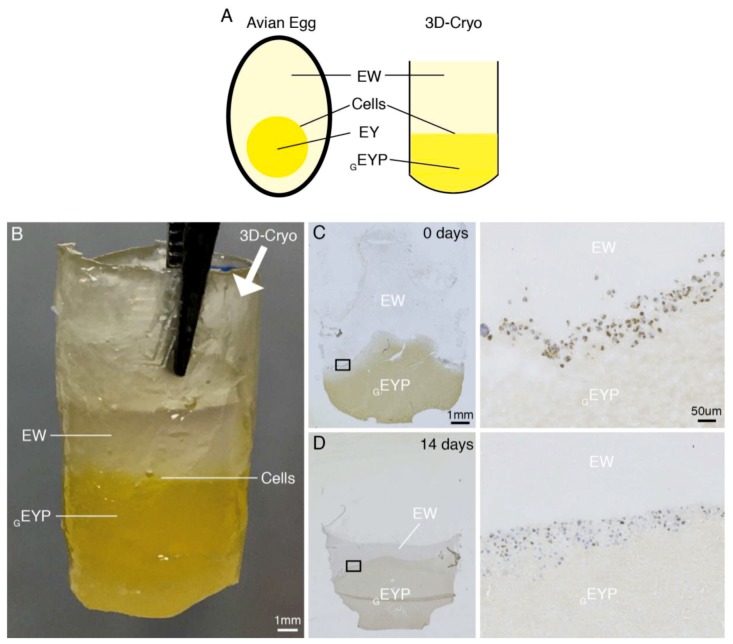
Proof of concept for _G_EYP + EW human in vitro model with 3D-Cryo well insert. (**A**) Planning of _G_EYP + EW experiment similar to the EY and EW interface around cells in avian development; (**B**) Picture of the well insert with both visible layers, _G_EYP and EW from the exterior; (**C**) T = 0: Sagittal section of 3D-Cryo well insert stained by IHC with anti-Ki-67 antibody. NS-SV-AC cells at the interface between _G_EYP and EW; (**D**) T = 14 days: Sagittal section of 3D-Cryo well insert stained by IHC with anti-Ki-67 antibody. NS-SV-AC cells at the interface between _G_EYP and EW. With panel (**C**,**D**), left pane is a general overview of the sagittal section at 25× and black box is the location of 200× magnigifcation on right. DAB stained Ki-67 positive cells are brown, and hematoxylin stained nuclei blue.

**Figure 5 materials-12-03480-f005:**
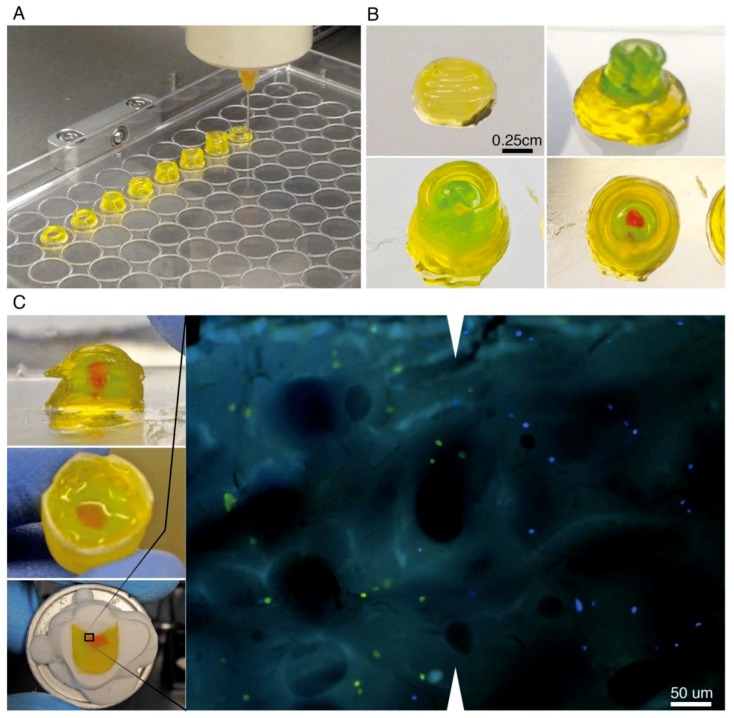
3D-Printing of freeze-thawed _G_EYP and implementation of 3D-Cryo well insert for characterization of extruded structure. (**A**) Pneumatic bioprinter print head rapidly extruding cylinders of freeze thawed _G_EYP on the lid of a 96 well plate; (**B**) Various layers involved in printing CAD designed model to study interfaces (ball and socket); in this structure, no cells are involved; (**C**) 3D-printed ball and socket model sectioned in half with glass coverslip to show sagittal/coronal plane (top left). Manually extruded reproduction of ball and socket model in histology well insert loaded with various fluorescent labelled cells (centre left). 3D-Cryo well insert mounted on cryotome cutting block sectioned to the core (bottom left). Examination of the resulting sagittal/coronal section from the 3D-Cryo well insert mounted on a glass slide and imaged with fluorescence light (Right). Cells are seen in green and blue distributed in the biomaterial. As they were extruded in separate inks, they are also separated in the images. White triangles show the interface line.

**Table 1 materials-12-03480-t001:** Brief summary on article’s figures.

Figure	Cell Type	Biomaterials	Tools	Test Parameter
1	NS-SV-AC	EYP + Media EYP + EW	- Chemical stain Sirius Red - 3D-Cryo well insert	- 0 & 14 days - Color of biomaterial - Cell distribution
2	NS-SV-AC HuSG-Fibro	EYP + Media EYP + EW Media	- IHC (Anti-Ki67) - 3D-Cryo well insert	- 0 & 14 days - Cell proliferation
3	N/A	EYP → _G_EYP	- Rotational rheometer - −20 °C Freezer	- 0, 1, 4, 7, 11, 15, 30 days - Storage Modulus (G’) - Loss Modulus (G’’)
4	NS-SV-AC	_G_EYP + EW	- IHC (anti-Ki67) - 3D-Cryo well insert	- 0 & 14 days - Cell distribution - Cell proliferation
5	NS-SV-AC (CFSE) NS-SV-AC (Hoeschst)	_G_EYP	- 3D-Extrusion printer - Glass syringe - 3D-Cryo well insert	- Design & extrudability - Structural Maintenance - Cell Localization

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
