# Peer review of "3D Cultures of Salivary Gland Cells in Native or Gelled Egg Yolk Plasma, Combined with Egg White and 3D-Printing of Gelled Egg Yolk Plasma"

_materials, 2019, doi:10.3390/ma12213480_

Round 1

Reviewer 1 Report

The authors of the present study examined the suitability of avian derived egg yolk plasma with or without the addition of egg white for tissue engineering applications including 3D-bioprinting. Additionally, they produced gelled egg yolk plasma by freeze-thaw gellation. As cell types they either used NS-SV-AC salivary cells or primary salivary fibroblasts. The experiments were analyzed by cryosectioning, chemical staining, IHC and rheological / mechanical measurements. Furthermore, the authors utilized 3D-Cryo well inserts which should help to maintain the native state of cells and biomaterial for (saggital) histological analyses.

Regrettably, the figures are missing in the submitted manuscript. Only the supplementary figures are available in a separate file. Because of this it is difficult to do a review of the present manuscript.

The introduction illustrates the background. Current literature is included and the aim of the study as well as the work for the present manuscript is clearly stated. At some points the introduction seems to be mixed up with the methods part. I would like to ask the authors to clearly separate this.

The subsequent materials and methods chapter is basically described in a good way. I would like to ask the authors to add city and country to the references for the used materials / devices.

For me it is not possible to review the results and discussion sections without the corresponding figures. In some supplementary figures the scale bars are missing. I would like to ask the authors to add these.

In general the authors showed an interesting approach to utilize avian egg biomaterials for use in tissue engineering and 3D-bioprinting. I would like to ask the authors to submit the missing figures in order to allow a review of the entire manuscript. Maybe it would also be beneficial to add a conclusion in order to point out the main findings.

Reviewer 2 Report

The paper entitled  ` 3D-Cultures and 3D-Bioprinting of Salivary Gland Cells in Native or Gelled Egg Yolk Plasma Combined with Egg White` contains some new and significant scientific information adequate to justify publication. However the reporting of the study need to be improved. The reporting of the experimental methods and results should be more complete and accurate. Abstract does not contain any quantitate data or clear results, only method.

Abstract opening sentence „Here we further investigate with a 3D-Cryo………….“, which sounds authors are continuing their  previous work without any description, which confuses the readers.

Abstract is full of abbreviation which make it pretty confusing to follow correct sequences. Please rework on abstract to make it easier to follow.

Since title of the papers highlights 3D printing and 3D culture as main focus of this study. In introduction, Literature part there should be more 3D printing/culture relevant references. what has been recently reviewed as novel biomaterials based approaches to improve 3D cell culture and 3D bioprinting in including seminal work of Cecchini et al (nat Histol Embryol. 2014 Jun;43(3):239-44) the murid glandular complex, composed of the submandibular and sublingual salivary glands (SSC). Please cite Pubmed ID PMID:23822094, PMID:31382208 and Appl. Sci. 2019, 9(4), 811; https://doi.org/10.3390/app9040811 in main text.

Please swap paragraph 1 and 2 since para 2 provides more valuable info about the bottleneck of the filed which this article fulfills.

The test parts made with parameters should be gathered in table format. Now text is quite hard to read.

About Mechanical characterization. Amount of samples is quite small. How much variation there was between measurements? What was the repeatability?

About the cell cultures, how many samples there were? Parallel and reference samples?

Reviewer 3 Report

This manuscript describe egg yolk plasma and egg white for 3D cell culuture. The cell culture materials from eggs are interesting. Some issues should be revised.
1. In introduction, a submitted manuscript is cited. I suggest some important results of that paper could be included in Supplementary.
2. Bioprinting should be changed to printing in title. Cell-laden printing can only be called bioprinting. Otherwise, it should be called 3D printing of biomaterials.
3. If the reported hydrogle can be claimed to be 3D printed, the basic printability analysis are needed. The authors are suggest to read some reports about analysis of hydrogel printability. "3D printing of complex GelMA-based scaffolds with nanoclay, Biofabrication,2019";“Research on the printability of hydrogels in 3D bioprinting,Scientific reports, 2016”

Round 2

Reviewer 1 Report

I would like to thank the authors for modifying the manuscript as well as for providing the respective figures. 

The manuscript improved a lot but I would like to ask the authors to exchange the term 3D-bioprinting by 3d-printing since they did not print their material together with cells (for example but not limited to line 199 and caption figure 5).

Apart from that the manuscript is fine for me and suitable for publication.

Reviewer 3 Report

OK.

Author Response

We thank the reviewer for their time in reviewing the manuscript a second time. This reviewer did not have any additional comments on the second review round.